# Local Administration of (−)-Epigallocatechin-3-Gallate as a Local Anesthetic Agent Inhibits the Excitability of Rat Nociceptive Primary Sensory Neurons

**DOI:** 10.3390/cells14010052

**Published:** 2025-01-05

**Authors:** Syogo Utugi, Risako Chida, Sana Yamaguchi, Yukito Sashide, Mamoru Takeda

**Affiliations:** Laboratory of Food and Physiological Sciences, Department of Life and Food Sciences, School of Life and Environmental Sciences, Azabu University, 1-17-71, Fuchinobe, Chuo-ku, Sagamihara 252-5201, Kanagawa, Japan; f22015@azabu-u.ac.jp (S.U.); f22001@azabu-u.ac.jp (R.C.); f22027@azabu-u.ac.jp (S.Y.); me2401@azabu-u.ac.jp (Y.S.)

**Keywords:** alternative medicine, extracellular single unit recording, lidocaine, (−)-epigallocatechin-3-gallate, trigeminal pain, primary afferent

## Abstract

While the impact of (−)-epigallocatechin-3-gallate (EGCG) on modulating nociceptive secondary neuron activity has been documented, it is still unknown how EGCG affects the excitability of nociceptive primary neurons in vivo. The objective of the current study was to investigate whether administering EGCG locally in rats reduces the excitability of nociceptive primary trigeminal ganglion (TG) neurons in response to mechanical stimulation in vivo. In anesthetized rats, TG neuronal extracellular single unit recordings were made in response to both non-noxious and noxious mechanical stimuli. Following the administration of EGCG, the mean firing rate of TG neurons to both non-noxious and noxious mechanical stimuli significantly decreased in a dose-dependent manner (1–10 mM), and both the non-noxious and nociceptive mechanical stimuli experienced the maximum suppression of discharge frequency within 5 min. These inhibitory effects lasted for approximately 20 min. These findings suggest that the local injection of EGCG into the peripheral receptive field suppresses the responsiveness of nociceptive primary sensory neurons in the TG, almost equal to that of the local anesthetic, 1% lidocaine. As a result, the local application of EGCG as a local anesthetic could alleviate nociceptive trigeminal pain that does not result in side effects, thereby playing a significant role in pain management.

## 1. Introduction

The sensory input from the orofacial region, transmitted by Aδ- and C-trigeminal ganglion (TG) neurons, is relayed to second-order neurons in the spinal trigeminal nucleus (SpV). Functionally, the SpV can be subdivided into three nuclei, ordered from rostral to caudal: oralis, interpolaris, and SpV caudalis (SpVc) [1,2,3]. In trigeminal primary and secondary neurons, two separate forms of nociceptive neurons exist: nociceptive-specific neurons and wide-dynamic-range (WDR) neurons [1,3]. Nociceptive-specific neurons respond uniquely to painful stimuli in their specific receptive fields, potentially transmitting information related to the location to higher brain regions [3]. In contrast, WDR neurons respond to noxious and non-noxious stimulation [1,2,3]. Increasing the intensity of nociceptive stimulation at the most sensitive part of the receptive field induces a corresponding increase in firing rate [1,2,3].

Pain management often involves the use of complementary alternative medicine (CAM), including herbal medicine and acupuncture, particularly when traditional Western medicine has failed or if there are worries about negative side effects [4,5,6,7]. Prior investigations have found that various dietary elements might affect defensive biological mechanisms present in the heart, nervous, and cancer defense systems [8,9]. For dental treatments, such as extracting teeth, patients are often given local anesthetics. Nevertheless, if local anesthetic blood levels become excessively elevated during the dental procedure, negative side effects, such as lightheadedness, visual disturbances, and auditory problems, may occur due to impacts on the central nervous system [5]. Consequently, there has been growing interest in recent times in using CAM to manage pain [10,11].

Catechins are a type of polyphenol and flavonoid which are naturally occurring dietary phytochemicals found in green tea. According to Mandel et al. [12], (−)-epigallocatechin-3-gallate (EGCG) is a key component of green tea and is the most active component, making up over 10% of the extract dry weight and 65% of the total catechin content. EGCG exhibits several biological and beneficial effects, such as antioxidant, anti-carcinogenic, and anti-inflammatory properties [13,14,15,16,17]. EGCG can influence neuronal excitability by altering voltage-gated Na (Nav), K (Kv), and Ca^2+^ (Cav) channels, acid-sensing ion channel (ASIC) 3, and non-selective ion channels, such as through the inhibition of glutaminergic synaptic transmission [18,19,20,21,22,23,24,25,26,27]. As noted by Kim et al. [24], under in vitro whole-cell voltage-clamp conditions, EGCG exhibits a dose-dependent inhibitory effect on both tetrodotoxin-resistant (TTX-R) and -sensitive (TTX-S) Nav channels in the dorsal root ganglion (DRG).

We recently observed that the localized injection of EGCG diminished the excitability of nociceptive neurons in the SpVc region [10]. This study also indicated that local half-dose injections of EGCG produced similar effects to those of a half-dose of the local anesthetic, lidocaine [10]. However, the investigation of local anesthetic effects from dietary ingredients affecting nociception was grounded in variations in the excitability of nociceptive secondary neurons, and not primary afferent activity. Consequently, it is yet to be ascertained if the local application of EGCG in rats diminishes the excitability of nociceptive primary neurons in response to noxious stimulation in vivo. Consequently, the objective of this study was to assess whether the acute local administration of EGCG in rats reduces the excitability of primary sensory neurons. Furthermore, we evaluated the effectiveness of EGCG versus a commonly used local anesthetic in terms of their ability to suppress trigeminal nociception. As a result, the application of EGCG as a local anesthetic could alleviate trigeminal nociceptive pain without causing side effects, thereby playing a significant role in pain management.

## 2. Materials and Methods

The Animal Use and Care Committee of Azabu University authorized all the experiments mentioned in this study. All experiments complied with the ethical rules of the International Association for the Study of Pain [28]. Maximum efforts were taken to reduce the number of animals used and to alleviate their suffering

### 2.1. Recording of Extracellular Single Unit Activity in Trigeminal Neurons

Male adult Wistar rats (210–260 g) were housed with a consistent light schedule (on 07:00–19:00). Room temperature was kept at 25 °C ± 1 °C. Food and water were provided freely. Electrophysiological data were collected from 15 rats. Each rat underwent anesthesia using 3% isoflurane along with a combination of anesthetic agents (0.3 mg/kg medetomidine, 4.0 mg/kg midazolam, 5.0 mg/kg butorphanol) and maintained with supplementary doses delivered through a cannula in the jugular vein, as needed. During the session, it was confirmed that the animal was properly anesthetized, as evidenced by no response to pinching its paw. The rectal temperature was kept at 37.0 °C ± 0.1 °C. Throughout the experiments, all wound edges were consistently treated with a local anesthetic, 2% lidocaine (Xylocaine). The animals were positioned in a stereotaxic frame, and a partial craniotomy was carried out 2–5 mm posterior to the bregma and 1–4 mm lateral to the central line. A tungsten microelectrode with an enamel coating and 3 MΩ impedance was carefully inserted into the cortex at a lateral distance of 2.5–3.5 mm from the midline and 2.5 to 3.5 mm posterior to the bregma (depth of 2.5–3.5 mm), as described previously [29]. The recording sites were identified according to a micromanipulator reading the distance from the bregma and the distance from the midline, and we plotted recording sites on the schematic diagram of the trigeminal ganglia, as described previously [29]. Next, the electrode was moved forward or backward by 10 μm increments with the help of a micromanipulator. Tungsten electrodes were used to extracellularly record the single unit activity in the TG region based on the stereotaxic coordinates of Paxinos and Watson [30]. The neural activity underwent amplification (DAM80; World Precision Instruments, Sarasota, FL, USA), filtration (0.3–10 KHz), monitoring with an oscilloscope (SS-7672; Iwatsu, Tokyo, Japan), and was recorded for subsequent analysis using Power Lab equipment Oxford, UK), as described previously [10,11,29].

### 2.2. Electrophysiological Recordings

The extracellular activity of single TG WDR neurons in response to the mechanical stimulation of the whisker pad was recorded and analyzed in the following way. To avoid causing sensitization to the peripheral mechanoreceptors, a paintbrush was quickly used to locate the general area of the receptive field. The next step involved mechanical stimulation using von Frey hairs (Semmes-Weinstein Monofilaments, North Coast Medical, USA) for 5 s at 5 s intervals [10,11,29]. The criteria for WDR neurons have been defined as follows: graded mechanical stimulation, both non-noxious and noxious, applied to the receptive field results in a proportional increase in firing frequency. Following the identification of nociceptive TG WDR neurons sensitive to the whisker pad, we determined the threshold for mechanical stimuli and the dimensions of the receptive fields. The mechanical response areas of neurons were identified using von Frey hair probes on the facial skin, after which they were traced onto a life-sized drawing of the face on tracing paper [10,11,29].

The mechanical stimulation-triggered TG neuronal discharges were measured by deducting the baseline activity from the stimulated activity. Spontaneous discharge frequencies were determined over 2–5 min. For each stimulus, post-stimulus histograms were constructed using 100 ms bins. The impact of subcutaneous EGCG (Sigma-Aldrich, Milano, Italy; 0.02 mL; 1 mM and 10 mM) and lidocaine (1% injection solution without epinephrine, lidocaine HCL, 2-Diethylamino-N-[2,6-dimethylpheny] acetamide; MW = 280.1; pH 5.0–7.0; equivalent to 37 mM, 0.02 mL, Aspen, Tokyo, Japan) into the peripheral receptive field (whisker pad) delivered via a Hamilton microsyringe was assessed at 5, 10, 15, 20, 30, and 40 min post-administration because the peak effect and recovery were thought to occur within this timeframe. In order to make a 10 mM stock solution, EGCG was dissolved in pure dimethyl sulfoxide. The stock solution was maintained at −20 °C until utilization. Just prior to use, the original solution was adjusted to the desired concentrations with saline. The mean spontaneous and mechanical stimulation-induced discharge rates, and the mechanical threshold before and after the subcutaneous administration of EGCG, were analyzed in the present study.

### 2.3. Data Analysis

Values are expressed as means ± SEM. A one-way repeated measures analysis of variance was utilized for statistical analysis, and subsequent Tukey–Kramer’s post hoc tests and Student’s t-test were performed on the behavioral and electrophysiological data. A p-value less than 0.05 was regarded as indicating a significant difference.

## 3. Results

### 3.1. Characteristics of TG Neurons That Innervate the Skin of the Face

In this study, the single unit extracellular activity was measured from 15 TG neurons. Ten TG neurons were subjected to subcutaneous EGCG injections to test their effects, whereas five neurons were evaluated for changes in neuronal excitability following lidocaine treatment. Figure 1A illustrates the receptive field of TG neurons that react to both non-noxious and noxious mechanical stimulation in the whisker pad. The majority of the recording sites were found in the maxillary branches of the TG (Figure 1B). No apparent variations were found in the placement of the recording sites between the EGCG and lidocaine groups. Figure 1C shows representative examples of TG neuronal unit discharge patterns. Applying mechanical stimulation at different levels to the receptive field resulted in an increase in the rate of neural firing of TG neurons in proportion to stimulus intensity. A total of 3 out of 15 neurons showed spontaneous discharges. Figure 1C (inset) displays characteristic action potential waveforms induced by mechanical stimulation. The mean threshold for spikes triggered by mechanical stimulation was 2.6 ± 1.1 g. All neurons observed were classified as WDR neurons [29] (Figure 1D). The administration of dimethyl sulfoxide locally showed no notable impact on TG neuronal responses to either non-noxious or noxious mechanical stimuli (n = 3) (Figure 1E), as described previously [29].

### 3.2. Impact of Localized EGCG Administration on TG Neuron Excitability in Response to Non-Noxious and Noxious Stimuli

The impact of a local subcutaneous injection of EGCG on the responsiveness of TG neurons to non-noxious mechanical stimulation is illustrated in Figure 2A. Five minutes after the subcutaneous administration of 10 mM EGCG into the receptive area, the TG neuronal response to non-noxious (2–10 g) mechanical stimulation was suppressed, activity returning to baseline levels in about 20 min. No significant alterations in the mechanical threshold, spontaneous activity, and size of the receptive field were detected following EGCG administration. Figure 2B illustrates the effects of EGCG on TG neuronal responses to non-noxious mechanical stimulation. The mean firing rates of TG neurons activated by non-noxious (6 and 10 g) mechanical stimulation significantly decreased following the injection of EGCG compared to before the injection (*p* < 0.05, n = 5), but they returned to normal levels after about 20 min.

Figure 2 further demonstrates representative examples of how a subcutaneous injection of 10 mM EGCG into the receptive field affects TG excitability in response to noxious mechanical stimulation. The EGCG injection led to a decrease in TG neuronal activity induced by noxious (15, 26, and 60 g) mechanical stimulation after 5 min, though the activity returned to the baseline control level within approximately 20 min (Figure 2A). The average firing rates of TG neurons triggered by noxious (15, 26, and 60 g) mechanical stimulation significantly declined following the EGCG injection in comparison to the control group (Figure 2B, *p* < 0.05; n = 5). EGCG significantly inhibited TG neuron firing induced by non-noxious (6 and 10 g) mechanical stimulation in a dose-dependent manner (Figure 3; 1 mM vs. 10 mM, *p* < 0.05, n = 5, each). The administration of EGCG resulted in a significant dose-dependent inhibition (Figure 3; 1 mM vs. 10 mM, *p* < 0.05, n = 5) of TG neuronal firing triggered by noxious (26 g and 60 g) mechanical stimuli.

### 3.3. The Effect of Noxious and Non-Noxious Stimuli on TG Neural Activity Post-EGCG Administration

We assessed the inhibitory impact of a 10 mM subcutaneous dosage of EGCG on reactions to both non-noxious and noxious stimuli. As shown in Figure 4, no notable difference was observed in the mean magnitude of inhibition discharge frequency induced by non-noxious and noxious stimuli with EGCG (n = 5). When comparing the percentage inhibition frequency of noxious stimulation-induced discharge by 10 mM EGCG between primary (TG) and secondary (SpVc) neurons, there is a significant difference (primary vs. secondary, 58% ± 3% vs. 71.6% ± 4.2%, *p* < 0.05) compared to the data from our previous study [10].

### 3.4. Examination of How EGCG and Lidocaine Influence TG Neuronal Response to Mechanical Stimulation

Finally, a comparison was made between the degree of inhibition of noxious stimulation-induced TG neuronal excitability by EGCG and lidocaine. The subcutaneous injection of 1% lidocaine into the receptive field typically affects the excitability of TG neurons in response to non-noxious and noxious mechanical stimulation, as shown in Figure 5A. The response of TG neurons to both non-noxious and noxious mechanical stimulation was diminished 10 min after lidocaine administration, with responses returning to the control levels within 45 min (Figure 5). Figure 5B provides an overview of the impact of lidocaine injection on TG neurons when exposed to non-noxious and noxious mechanical stimulation, showing discharge frequencies significantly returned to control levels within 45 min. The average inhibition of TG nociceptive transmission was comparable between 10 mM EGCG and 37 mM lidocaine, with the average inhibition levels being equivalent (37 mM lidocaine vs. 10 mM EGCG: non-noxious, 56.7% ± 6.2% vs. 50.0% ± 1.5%, n = 5, NS; noxious, 58.3% ± 3.1% vs. 51.9% ± 2.4%, n = 5, NS) (Figure 6).

## 4. Discussion

### 4.1. Topical Application of EGCG Reduces the Excitability of Nociceptive Primary TG Neurons

We have recently shown that the local application of EGCG reduces the excitability of tri geminal nociceptive secondary neurons in vivo [10]. However, this finding was based on nociceptive secondary neuronal activity and not nociceptive primary afferent activity. Consequently, the purpose of this study was to examine if applying EGCG locally to rats reduces the responsiveness of TG neurons to both nociceptive and non-nociceptive mechanical stimuli in vivo without the presence of inflammatory or neuropathic pain.

The primary results of the current research are that (i) the local injection caused a dose-dependent decrease in the mean firing rate of TG neurons in response to both non-noxious and noxious mechanical stimuli (1–10 mM); (ii) the inhibitory effect of EGCG on the discharge frequency in reaction to both non-noxious and noxious mechanical stimuli was reversible (within 15–20 min); and (iii) there was no significant impact of the local vehicle injection on TG neuronal activity triggered by non-noxious or noxious mechanical stimulation.

Previous reports indicate that administering 0.1 mM EGCG notably reduces Nav cur rents elicited by a step-pulse protocol in DRG neurons [24]. Moreover, in preparations of brain slices, the use of 0.01 mM EGCG results in the hyperpolarization of membrane potentials and diminishes the spontaneous firing rates of vestibular neurons [21]. We hypothesized that once administered locally in vivo, 10 mM EGCG solution will be diluted in the extracellular fluid, resulting in an approximate concentration of 100 μM (0.1 mM), and therefore we applied a concentration of 10 mM EGCG to evaluate the discharge frequency of TG neurons induced by mechanical stimulation. In this analysis, we explored how the direct application of EGCG influenced the discharge frequencies of an extracellular single unit in the primary sensory neurons of the trigeminal ganglion. Our findings, therefore, provide the first evidence that the local administration of EGCG in the peripheral receptive area reduces the excitability of “primary” sensory neurons, potentially via the blockade of Nav channels and the facilitation of Kv channels in the nociceptive nerve terminals of the TG. However, validation studies, such as in vitro patch clamp experiments on dissociated TG neurons, are required to confirm this.

### 4.2. Local Administration of EGCG Diminishes TG Neuronal Excitability Through Peripheral Mechanisms

It is widely recognized that the pathway of nociceptive sensory transmission relies on four general processes: (i) transduction occurs when the peripheral terminal transforms external stimuli; (ii) the generation and initiation of action potentials; (iii) propagation along the axon, responsible for transmitting action potentials; and (iv) transmission, in which the central terminal serves as the presynaptic element [3,31].

In this study, we found that administering a local injection of EGCG directly into the peripheral receptive field in vivo elucidates the responsiveness of nociceptive primary sensory TG neurons. Since EGCG suppresses ASIC3 currents (mechanosensitive channel candidate) in Chinese hamster ovarian cells [26], it is probable that the local application of EGCG acts via ASICs, as a potential candidate for mechanoreceptors in the nerve terminal of TG neurons. Thus, the local administration of EGCG effectively inhibits generator potentials and, as a consequence, dampens the action potential firing of the nociceptive TG neurons.

The Nav channels responsible for transmitting action potentials related to pain perception are typically categorized into two groups. It appears that nociceptive DRG neurons, usually small to medium in size, selectively express Nav channels that are not affected by TTX, while TTX-S Nav channels are represented in Aβ-/Aδ-(large-/medium-sized) sensory neurons [3,32]. As for the effect of EGCG on primary sensory neurons, according to Kim et al. [24], EGCG inhibited the activity of both TTX-R and TTX-S Nav channels in DRG neurons, with the effect being both dose-dependent and reversible in an in vitro setup. In the current study, we observed a decrease in discharge frequency in reaction to both non-noxious and noxious mechanical stimuli due to EGCG. There was no significant difference in the mean inhibition effect of EGCG on the TG neuronal discharge frequency between noxious and non-noxious stimuli. Taken together, a local injection of EGCG likely decreases the excitability of TG neurons through the inhibition of TTX-R and TTX-S Nav channels in the nociceptive nerve terminals.

Kv channels play a crucial role in the nervous system, participating in various essential functions, such as establishing the resting membrane potential, determining the action potential threshold, contributing to the repolarization of excitable cells, and neurotransmitter release through channels that exhibit slow-inactivating sustained (K-current) and fast-inactivating transient (A-current) properties [3,33]. EGCG is known to stimulate Kv channels, resulting in vasodilation associated with vascular smooth muscle activity [27] by opening Kv channels and reducing the spontaneous firing rates of acutely dissociated vestibular nuclear neurons in vitro [21]. When considered together, these results imply that administering EGCG locally in the peripheral receptive field dampens the excitability of TG neurons that respond to mechanical stimulation, perhaps through the triggering of Kv channels in both non-nociceptive and nociceptive nerve terminals associated with TG neurons. However, additional introductory studies are required to investigate this possibility.

EGCG has been shown to influence various voltage-gated ion channels, with studies indicating that it inhibits the activity of cardiac L-type calcium channels in both vascular smooth muscle and cardiac muscle [23,25]. Sensory neurons with diameters ranging from 15 to 30 µm (small) and 31 to 40 µm (medium) predominantly express T-type Cav channels [34,35], and these neurons can be categorized as unmyelinated C-neurons and myelinated Aδ-neurons, respectively [36]. Jevtovic-Todorovic and Todorovic [37] found that increasing the amplitude of T-type Cav currents leads to a decrease in the excitability threshold, thereby enhancing neuronal responsiveness and consequently an increase in the probability of neuron burst-firing. Collectively, these findings indicate that EGCG might suppress the magnitude of T-type calcium currents, thereby potentially lowering the activity of the TG neuronal firing rate in response to noxious mechanical stimuli.

### 4.3. Importance of EGCG in Reducing Nociceptive Stimulation Effects on TG Neurons

There is increasing interest in using CAM for the treatment of both acute and chronic pain [7,38,39,40]. Patients frequently turn to CAM to find relief from their pain when conventional Western treatments are ineffective [6]. Administering local anesthesia may lead to adverse effects on both the central nervous system and the cardiovascular system [41]. As such, the notion of developing and recognizing new dietary ingredients that can serve as pain relief medications without adverse effects is well founded. In this study, we analyzed the average extent of the inhibition of TG neuronal excitability between EGCG and the Nav channel blocker, 1% lidocaine. The average level of inhibition on SpVc neuronal discharge frequency was nearly the same for EGCG (10 mM) and 1% lidocaine, suggesting that the effectiveness of the inhibitory effects of nociceptive transmission by EGCG was four-fold higher than 1% lidocaine. The findings in these trigeminal first-order neurons were consistent with previous findings in trigeminal second-order neurons [10]. Additionally, this demonstrated the validity of second-order neuron activity as a method for discovering the previous local anesthetic effect of a phytochemical, such as genistein, previously identified in trigeminal second-order neurons [11].

When the inhibition rates of EGCG for primary neurons and second-order neurons were compared, the inhibition rate of second-order neurons was relatively greater in both non-noxious and noxious stimuli, as described in the Results Section 3.3, compared to data from our previous study [10]. Sensory information from peripheral receptive fields is conveyed to second-order neurons by primary sensory neurons. Therefore, the information does not converge at the primary neuron level but second-order sensory neurons receive convergence from multiple primary neurons and interneurons [3]. Although the exact difference in the inhibition rate of the discharge frequency in response to EGCG-induced noxious stimuli is unclear, it is presumed that the difference in the convergence rate between primary and second-order neurons is a major factor. The fact that EGCG suppresses neuronal discharge frequency more strongly in the second order than in the first order is interesting because it suggests that a higher center in the pain transmission pathway is involved in increasing the rate of pain relief. However, additional research is required to clarify this possibility.

## 5. Conclusions

The current study is the first to demonstrate that the local application of EGCG to the peripheral receptive area reduces the responsiveness of nociceptive primary sensory neurons found in the TG, potentially through the suppression of ASICs and Nav channels and opening Kv channels in the nociceptive nerve terminals (Figure 7). Consequently, using EGCG, a phytochemical, as a local anesthetic could offer relief from trigeminal nerve pain with minimal side effects, thereby promoting the utilization of CAM.

## Figures and Tables

**Figure 1 cells-14-00052-f001:**
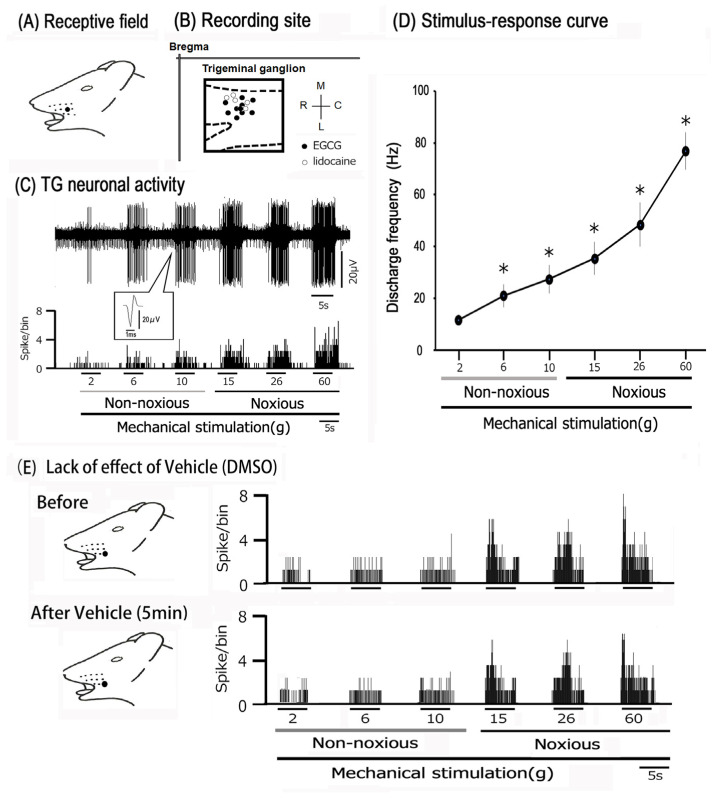
Overview of TG neuron activity patterns in reaction to mechanical stimulation of facial skin. (**A**) The area of the facial skin activated by the whisker pad (shaded region). (**B**) Distribution of TG neurons activated by non-noxious and noxious mechanical stimuli on the facial skin (n = 15). The inset illustrates an example of how TG recording sites are identified (2–5 mm posterior to bregma and 1–4 mm lateral to midline). R, rostral; C, caudal; M, medial; L, lateral. Trigeminal nerve (I, II, and III). (**C**) An example of firing in SpVc WDR neurons, triggered by both non-noxious (2, 6, and 10 g) and noxious (15, 26, and 60 g) mechanical stimulation of the orofacial skin. Upper trace: TG neuronal activity; lower trace: post-stimulus histogram. Inset: a representative waveform of an action potential triggered by mechanical stimulation. (**D**) SpVc WDR neuron stimulus–response characteristics (n = 15) * *p* < 0.05 for comparison of 2 g vs. 6 g, 10 g, 15 g, 26 g, and 60 g. The values are mean ± standard error. (**E**) Effect of subcutaneous administration of vehicle (DMSO) in the peripheral receptive field on the TG neuronal activity induced by non-noxious and noxious mechanical stimulus.

**Figure 2 cells-14-00052-f002:**
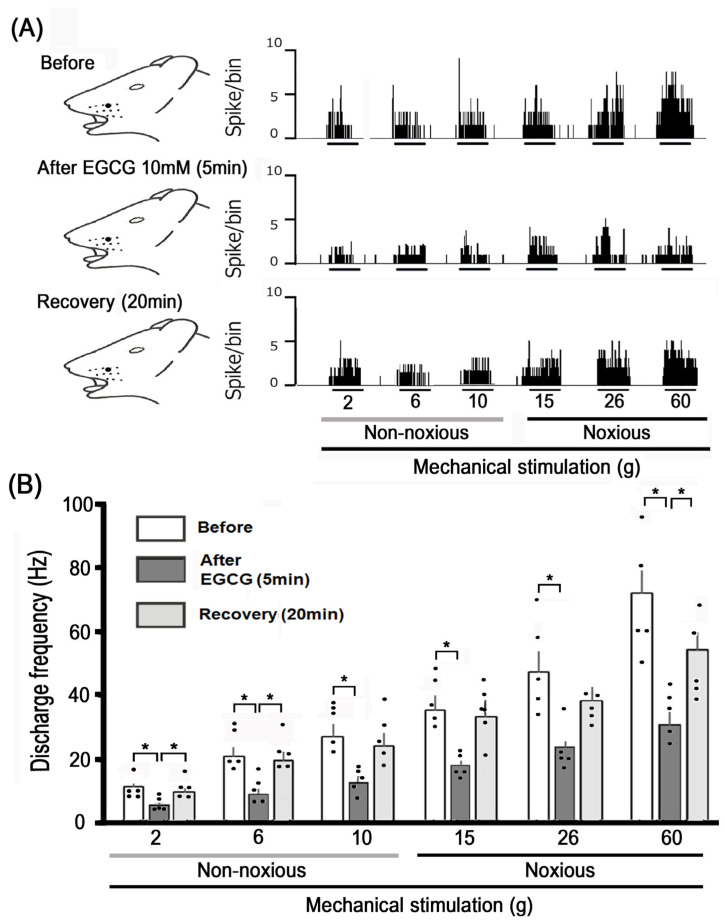
Subcutaneous administration of EGCG in the peripheral receptive field affects TG neuronal activation induced by non-noxious and noxious mechanical stimulus. (**A**) Representative cases of TG neuronal activity elicited by non-noxious (2, 6, and 10 g) and noxious (15, 26, and 60 g) mechanical stimuli before and 10 min and 20 min post-administration. (**B**) Temporal pattern of EGCG application in the peripheral receptive field on the average firing rate of TG neurons; response to non-noxious and noxious mechanical stimulation. * *p* < 0.05, compared with 5 min after EGCG administration (n = 5). The values are mean ± standard error.

**Figure 3 cells-14-00052-f003:**
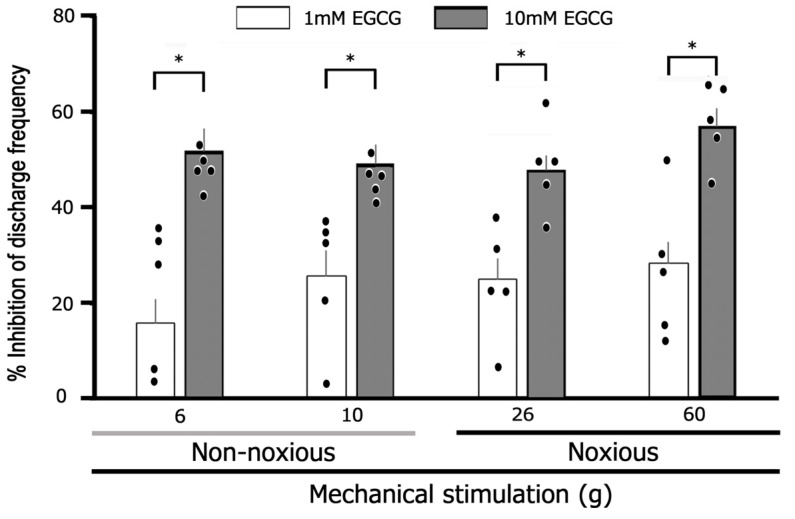
EGCG elicited a dose-dependent reduction in the mean firing frequency of TG neurons subjected to both non-noxious and noxious mechanical stimuli. * *p* < 0.05, 1 mM (n = 5) vs. 10 mM EGCG (n = 5). The values are mean ± standard error.

**Figure 4 cells-14-00052-f004:**
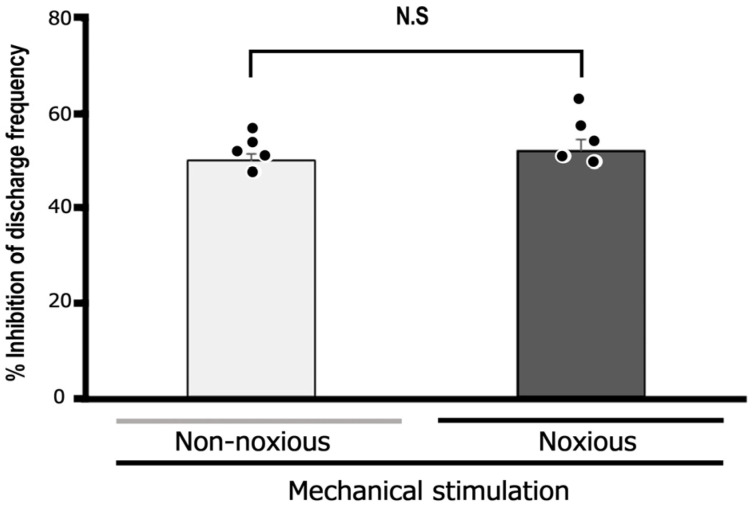
Evaluation of the decrease in TG neuron discharge frequency induced by EGCG between non-noxious and noxious stimuli. Non-noxious vs. noxious stimulation (n = 5). NS, not significant. The values are mean ± standard error.

**Figure 5 cells-14-00052-f005:**
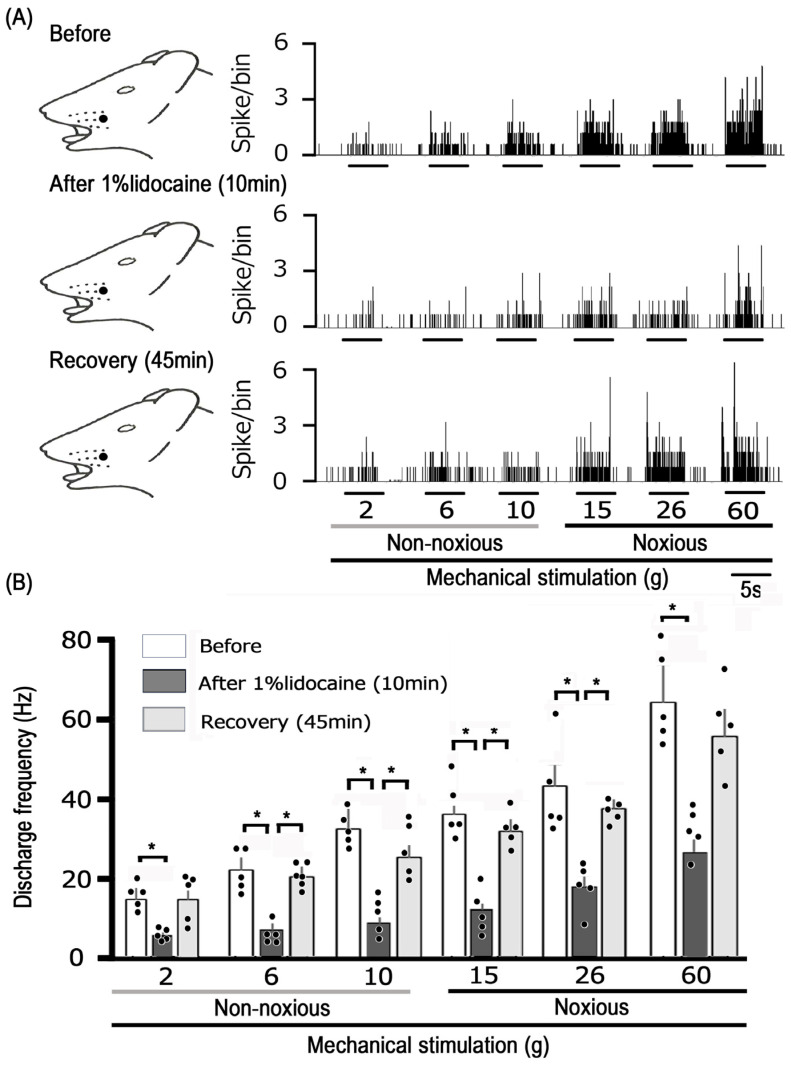
The consequences of administering 1% lidocaine (37 mM) subcutaneously in the peripheral receptive field on TG neuron responses to non-noxious and noxious stimuli. (**A**) Typical examples of TG neuron activity in response to non-noxious (2, 6, and 10 g) and noxious (15, 26, and 60 g) mechanical stimulation before and 10 min and 45 min after 1% lidocaine administration (37 mM). Receptive field of the whisker pad in the facial skin. The shaded region represents the location and extent of the receptive field. (**B**) Temporal pattern of lidocaine application in the peripheral receptive field on the average firing rate of TG neurons responding to non-noxious and noxious mechanical stimulation. * *p* < 0.05, compared with 10 min after 1% lidocaine administration (n = 5). The values are mean ± standard error.

**Figure 6 cells-14-00052-f006:**
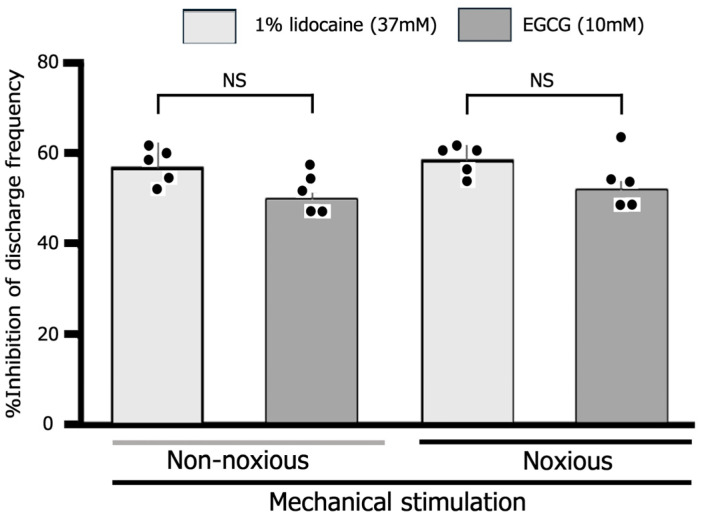
Comparison between 1% lidocaine and EGCG of mean magnitude of TG neuron inhibition in response to non-noxious and noxious mechanical stimulation. NS, not significant, lidocaine, n = 5; EGCG, n = 5. The values are mean ± standard error.

**Figure 7 cells-14-00052-f007:**
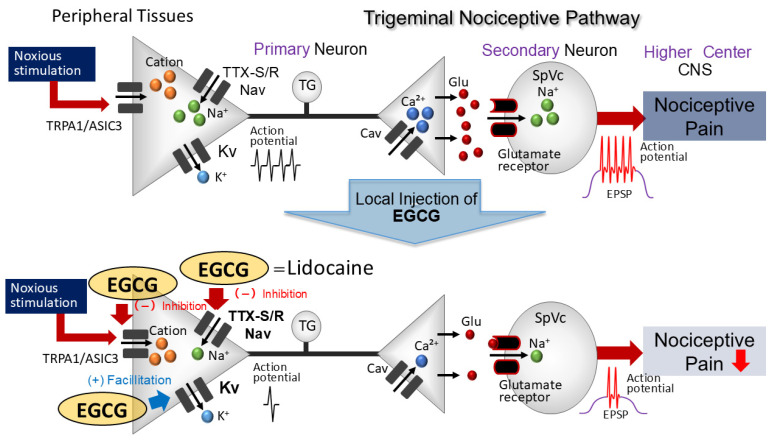
A possible mechanism underlying EGCG-induced inhibition of TG neuronal discharge in response to nociceptive mechanical stimulation. EGCG, when administered locally to peripheral tissues, inhibits the formation of both generator potentials and action potentials in the peripheral endings after nociceptive stimulation via the inhibition of mechanosensitive ionic channels (acid-sensing ion channel [ASIC] and transient receptor potential ankyrin 1 [TRPA1]), tetrodotoxin-resistant (TTX-R) and -sensitive (TTX-S) voltage-gated sodium (Nav) channels, and the facilitation of voltage-gated potassium (Kv) channels. EGCG’s potency is almost equivalent to that of Nav channel blockers, such as the commonly utilized local anesthetic lidocaine Cav, voltage-gated calcium; TG, trigeminal ganglion; and SpVc, trigeminal spinal nucleus caudalis.

## Data Availability

Data presented in this study are available on request from the corresponding author.

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
