# Peer review of "Local Administration of (−)-Epigallocatechin-3-Gallate as a Local Anesthetic Agent Inhibits the Excitability of Rat Nociceptive Primary Sensory Neurons"

_cells, 2025, doi:10.3390/cells14010052_

Round 1

Reviewer 1 Report

Comments and Suggestions for Authors

In this report, Utugi and colleagues examined how the local administration of EGCG in rats would affect primary trigeminal ganglion neuron excitability. The authors used extracellular single-unit recordings and measured the response to both nonnoxious and noxious mechanical stimuli. The authors further compared these responses to those obtained with the local application of 1% lidocaine. Additionally, the authors tested whether the subcutaneous administration of EGCG would affect the mechanical stimulation of both nonnoxious and noxious stimuli.

Overall, this is a well-written paper and carefully explained. However, there are some concerns that the authors need to address:

1. It is stated that DMSO alone did not notably affect the TG responses (lines 194-196). However, the authors did not provide the raw data. This data should be provided and shown in Figure 1.

2. The authors indicate that EGCG was administered subcutaneously. However, they do not indicate where exactly on the animal was this performed. Please provide this information.

3. The bar graphs should also be scatter plots so that the reader can see the scatter of the data. The figure legends should state the values are mean +/- SE. It is in the methods, but it should also be included in the figure legends.

Author Response

Reviewer #1: 

In this report, Utugi and colleagues examined how the local administration of EGCG in rats would affect primary trigeminal ganglion neuron excitability. The authors used extracellular single-unit recordings and measured the response to both nonnoxious and noxious mechanical stimuli. The authors further compared these responses to those obtained with the local application of 1% lidocaine. Additionally, the authors tested whether the subcutaneous administration of EGCG would affect the mechanical stimulation of both nonnoxious and noxious stimuli. Overall, this is a well-written paper and carefully explained. However, there are some concerns that the authors need to address:

Answer: We appreciate Reviewer’s careful reviews on our manuscript. We have carefully checked reviewer’s constructive and helpful comments and made necessary corrections to revised manuscript (marked by red font) We believe that your comments helped to improve our manuscript.

According to editor’s suggestion, concerning the similarity rate, we had revised the style of the parts of the main text that overlap with previous papers (marked by blue font) in revised manuscript, as much as possible. 

Comment 1: It is stated that DMSO alone did not notably affect the TG responses (lines 194-196). However, the authors did not provide the raw data. This data should be provided and shown in Figure 1.

Answer 

We greatly appreciate the reviewer’s careful review on our manuscript. We already previously reported that no significant effect of vehicle (DMSO) injection into the receptive field on the TG WDR neuronal activity responding to mechanical stimulation in the previous studies (Toyota et al., 2023, J Pain 24:540-549; Sashide et al., 2024, J Pain,25:755-765). Therefore, we did not show that vehicle control data in this manuscript. However, we have revised Results 3.2, as follows (page 6) ; Administration of dimethyl sulfoxide locally showed no notable impact on TG neuronal responses to either non-noxious or noxious mechanical stimuli (n = 3, data not shown), as described previously [29].

Comment 2: The authors indicate that EGCG was administered subcutaneously. However, they do not indicate where exactly on the animal was this performed. Please provide this information.

Answer 

We greatly appreciate the reviewer’s careful review on our manuscript. According to reviewer’s comment, we have revised Material and Methods, 2.2 as follows (page 3); The impact of subcutaneous EGCG (Sigma-Aldrich, Milano, Italy; 0.02 mL; 1 mM and 10 mM) and lidocaine (1% injection solution without epinephrine, Lidocaine HCL, 2-Diethylamino-N-[2,6-dimethylpheny] acetamide; MW = 280.1; pH 5.0-7.0; equivalent to 37 mM, 0.02 mL, Aspen Japan), into the peripheral receptive field (whisker pad) delivered via a Hamilton microsyringe was assessed at 5, 10, 15, 20, 30, and 40 minutes post-administration because the peak effect and recovery were thought to occur within this timeframe.

Comment 3: The bar graphs should also be scatter plots so that the reader can see the scatter of the data. The figure legends should state the values are mean +/- SE. It is in the methods, but it should also be included in the figure legends.

Answer 

We greatly appreciate the reviewer’s careful review on our manuscript. According to reviewer’s comment, we have added the scatter of the data in each bar graph and revised the figures 2, 3, 4, 5 and 6. We have added following sentences in the revised figures 2, 3, 4, 5 and 6 of legends.; the values are mean ± standard error 

Reviewer 2 Report

Comments and Suggestions for Authors

The authors aimed to expand their study about the effects of EGCG on nociceptive neurons and concluded that EGCG could decrease the excitability of TG neurons. However, there are some problems of the manuscript:

1. The name of the recorded neurons was confused, WDR neurons or nociceptive-specific neurons should be clarified.

2. If the anesthetics affected the neuronal excitability if they were not  continuously perfused?

3. Did EGCG and lidocaine affect the receptive field of these neurons?

4. How the recording sites were identified? A histological method should be provided.

5. In Figure 2, the firing rate to non-noxious stimulus 2 g was clearly decreased by EGCG.

6. PBS rather than DMSO should be used as vehicle for comparison, and at least 3 doses should be study if the authors claimed dose-dependent?

7. The baseline of neuronal firing in Figure 2 for EGCG and 5 for lidocaine were significantly different.

8. It did make sense to compare the inhibition of drugs on non-noxious and noxious stimuli-induced firing rate.

Comments on the Quality of English Language

The language requires to be improved. There are too many repetition of results description. For example, Line 170-172 and Line 175-177.

Author Response

Reviewer #2:

The authors aimed to expand their study about the effects of EGCG on nociceptive neurons and concluded that EGCG could decrease the excitability of TG neurons. However, there are some problems of the manuscript:

Answer: We appreciate Reviewer’s careful reviews on our manuscript. We have carefully checked reviewer’s constructive and helpful comments and made necessary corrections to revised manuscript (marked by red font) We believe that your comments helped to improve our manuscript.

According to editor’s suggestion, concerning the similarity rate, we had revised the style of the parts of the main text that overlap with previous papers (marked by blue font) in revised manuscript, as much as possible.

Comment 1: The name of the recorded neurons was confused, WDR neurons or nociceptive-specific neurons should be clarified.

Answer

We greatly appreciate the reviewer’s careful review on our manuscript. In this study, we recorded and analyzed only WDR neurons. Therefore, we have revised the Material s and Methods section 2.2. as follows (page 3); The extracellular activity of single TG WDR neurons in response to mechanical stimulation of the whisker pad was recorded and analyzed in the following way.

Comment 2. If the anesthetics affected the neuronal excitability if they were not continuously perfused?

Answer

We greatly appreciate the reviewer’s careful review on our manuscript. We described in the method section Page 2); Each rat underwent anesthesia using 3% isoflurane along with a combination of anesthetic agents and maintained with additional doses through a cannula in the jugular vein, as required. During the session, it was confirmed that the animal was properly anesthetized, as evidenced by no response to pinching its paw. In this study, we did not observe any significant changes in discharge frequency responding to mechanical stimulation before and after additional administration of anesthetic agents.

Comment 3. Did EGCG and lidocaine affect the receptive field of these neurons?

Answer

We greatly appreciate the reviewer’s careful review on our manuscript. In this study we did not observe change in the receptive field size before and after both EGCG and lidocaine injections. Therefore, we have revised result section, 3.2. as follows (page 5); No significant alterations in the mechanical threshold, spontaneous activity and size of receptive field were detected following EGCG administration.

Comment 4. How the recording sites were identified? A histological method should be provided.

Answer

We greatly appreciate the reviewer’s careful review on our manuscript. We have revised Material and Method sections 2.1, as follow; An enamel-coated tungsten microelectrode, having 3 MΩ impedance, was meticulously advanced into the cortex at a lateral distance of 2.5-3.5 mm from the midline and 2.5 to 3.5mm posterior to bregma (depth of 2.5-3.5 mm), as described previously [29]. The recording sites were identified according to micromanipulator reading of the distance from the bregma and the distance from the midline, we  plotted recording sites on the schematic diagram of trigeminal ganglia, as described previously [29].  

Comment 5. In Figure 2, the firing rate to non-noxious stimulus 2 g was clearly decreased by EGCG.

Answer

We greatly appreciate the reviewer’s careful review on our manuscript. In this study, we observed that the reversible inhibition of local injection of EGCG on the TG neuronal discharge frequency responding mechanical stimulation. Thus, we have revised the typical example data in revised Figure 2A (please see revised figure 2)

Comment 6. PBS rather than DMSO should be used as vehicle for comparison, and at least 3 doses should be study if the authors claimed dose-dependent ?

Answer

We greatly appreciate the reviewer’s careful review on our manuscript. In our previous study, we observed that both DMSO and saline have no significant effect on the TG neuronal discharge frequency. In this study,

We found that following administration of EGCG, the mean firing rate of TG neurons to both non-noxious and noxious mechanical stimuli significantly decreased in a dose-dependent manner (1-10 mM). Thus, we believed that EGCG inhibits the TG neuronal discharge frequency responding mechanical stimulation in concentration-dependent manner.

Comment 7. The baseline of neuronal firing in Figure 2 for EGCG and 5 for lidocaine were significantly different.

Answer

We greatly appreciate the reviewer’s careful review on our manuscript. Since the number of spontaneous discharges of recorded neurons varied, the lidocaine and EGCG data were replaced with data of the same frequency. We have replaced typical example data in revised Figure 2A )

Comment 8. It did make sense to compare the inhibition of drugs on non-noxious and noxious stimuli-induced firing rate.

Answer

We greatly appreciate the reviewer’s careful review on our manuscript. The objective of the current study was to investigate whether administering EGCG locally in rats reduces the excitability of nociceptive primary TG WDR neurons in response to non-noxious and noxious mechanical stimulation in vivo. In this study, in order to investigate whether the response to noxious and non-noxious stimuli was suppressed as with lidocaine, the inhibition rates for non-noxious and noxious stimuli were compared.

Comments on the Quality of English Language

The language requires to be improved. There are too many repetition of results description. For example, Line 170-172 and Line 175-177.

Answer

We greatly appreciate the reviewer’s careful review on our manuscript. There are two statements below, but the former describes the effect of EGCG on representative neurons, and the latter describes the average value of multiple cases and whether or not there was statistical significance. These two statements indicate different meanings, and we believe them to be necessary statements.

Line 170-172

Five minutes after the subcutaneous administration of 10 mM EGCG into the receptive area, the TG neuronal response to non-noxious (2-10 g) mechanical stimulation was suppressed, with activity returning to control levels within approximately 20 min.

Line 175-177.

The mean firing rates of TG neurons activated by non-noxious (6 and 10 g) mechanical stimulation significantly decreased following the injection of EGCG compared to before the injection (p < 0.05, n = 5), and returned to control levels within 20 min.

Round 2

Reviewer 1 Report

Comments and Suggestions for Authors

I asked the authors to submit the raw data to show that DMSO did not affect TG responses. In their reply, they basically ignored the critique. The authors MUST show the raw data. It is an easy fix or are they hiding something? It is pure arrogance that they chose to not submit the data asked from them. Until that raw data is submitted, then this paper should not be accepted in its present form.

Author Response

Reviewer #1:

Comments and Suggestions for Authors

I asked the authors to submit the raw data to show that DMSO did not affect TG responses. In their reply, they basically ignored the critique. The authors MUST show the raw data. It is an easy fix or are they hiding something? It is pure arrogance that they chose to not submit the data asked from them. Until that raw data is submitted, then this paper should not be accepted in its present form.

Answer

We greatly appreciate the reviewer’s careful review on our manuscript. We apologize for my previous response being inappropriate. According to reviewer’s suggestion. We have added data for vehicle (DMSO) did not affect TG responses in the revised Figure 1(E). We have revised the concerning description as follows (Page 5); Administration of dimethyl sulfoxide locally showed no notable impact on TG neuronal responses to either non-noxious or noxious mechanical stimuli (n = 3)(Figure 1E), as described previously [29].

Also We have added figure 1E legend, as follows;(E) Effect of subcutaneous administration of vehicle (DMSO) in the peripheral receptive field on the TG neuronal activity induced by non-noxious and noxious mechanical stimulus.

According to editor’s suggestion, concerning similarity rate, we have corrected English expressions as much as possible in the duplicated parts of the paper, except for the parts where technical terms cannot be replaced (marked by blue font). Please note that the rest of the book contains words and phrases that cannot be replaced by neurophysiology jargon and cannot be rewritten.

Reviewer 2 Report

Comments and Suggestions for Authors

The manuscript was improved greatly, there are only some small problems:

1. Use same past tense for whole manuscript. 

2. The recording sites can be histologically marked by electrical lesion via the recording electrode in future study.

Author Response

Comments and Suggestions for Authors

The manuscript was improved greatly, there are only some small problems:

  1. Use same past tense for whole manuscript. 

Answer

We greatly appreciate the reviewer’s careful review on our manuscript. We have checked and revised same past tense for whole manuscript as much as possible. 

  1. The recording sites can be histologically marked by electrical lesion via the recording electrode in future study.

Answer

We greatly appreciate the reviewer’s careful review on our manuscript. Thank for informative suggestion. We are planning to marked by electrical lesion via the recording electrode in future study.

According to editor’s suggestion, concerning similarity rate, we have corrected English expressions as much as possible in the duplicated parts of the paper, except for the parts where technical terms cannot be replaced (marked by blue font). Please note that the rest of the book contains words and phrases that cannot be replaced by neurophysiology jargon and cannot be rewritten.

Round 3

Reviewer 1 Report

Comments and Suggestions for Authors

The authors have now addressed my concerns.